# Cross-View Yaw Estimation in Location Uncertainty with Line-Aligning Yaw Scoring

## Abstract

Accurate rotation estimation is crucial in autonomous navigation and AR/MR (Augmented/Mixed Reality) applications. Small angular errors can lead to significant misalignment or navigation failures. Among the three rotation angles—pitch, roll, and yaw—yaw is the most challenging to estimate, as it lacks direct geometric cues, such as gravity-aligned structures. Yaw estimation given a BEV (Bird's Eye View) image is treated as an inseparable cross-view localization problem that accompanies location and inevitably hypothesizes the height and distance of the ground pixel. We introduce LAYS, a line-alinging yaw scoring approach that enables precise yaw estimation. We propose a 3D voting-based search that effectively isolates the 1-DoF yaw component, enabling robust estimation without relying on ground-truth position or assuming ground height. In our method, BEV pixels are matched to a ground view column based on feature similarity. Using the relative yaw of the ground column, match scores are assigned to a yaw bin for each 2D pose pixel. To address location uncertainty, our method identifies line correspondence between the ground and BEV, and formulates the problem such that one such correspondence is sufficient to determine yaw. LAYS achieves state-of-the-art sub-degree yaw accuracy, improving from 6.55% to 34.81% on the Mapillary Geo-Localization, 41.36% to 67.05% on the Ford Multi-AV, 3.14% to 45.51% on the KITTI, and 12.39% to 23.67% on the VIGOR dataset, setting a new benchmark for precise localization in real-world scenarios.

## 1 Introduction

Accurate global localization is essential for autonomous navigation, robotics, and AR/MR systems. Orientation error, in particular, leads to severe navigation errors or visual misalignment. Among the three rotation angles, yaw cannot be estimated from common gravity-aligned cues. This makes yaw a persistent bottleneck: without accurate yaw, global orientation alignment collapses into an error-prone joint estimation of translation and rotation.

Although cross-view localization has advanced, yaw is treated as a byproduct of translation or assumed to be approximately known, leaving yaw estimation underexplored and underperforming. Cross-view localization estimates are posed in global coordinates, matching ground-level views with a BEV (Bird's Eye View), typically using satellite imagery (Shi et al., 2019b; Shi & Li, 2022). Recent vision methods are applied for the setup (Song et al., 2023); however, domain-specialized approaches are widely explored to address the severe gap between the two image domains and the temporal scene changes. They tightly couple location and rotation, assuming flat ground (Shi & Li, 2022; Lentsch et al., 2023; Wang et al., 2024) or hypothesize height (Xia & Alahi, 2025) to map two views. Real-world environments introduce ambiguities, on slopes, or in urban clutter that projection-based methods cannot resolve.

We argue that yaw should be treated as an independent and tractable subproblem rather than a residual of full 3-DoF estimation, and reformulate it as a standalone 1-DoF problem. Yaw can be estimated from a single correspondence between the gravity-aligned view and the BEV, given the location; however, an accurate position is not commonly available. In such uncertainty, one *line correspondence* between two views, a ground column and a BEV radial line, is sufficient to estimate yaw. From this observation, we introduce **LAYS(Line-Aligning Yaw Scoring)**, treating cross-view yaw estimation as a *line alignment problem* rather than a pixel-level correspondence problem. LAYS ex-

Figure 1: Yaw estimation from relative yaw and column-pixel matching. **(a)** Assume the BEV pixel is matched to the ground column, corresponding to a radial line the pixel is on. **(b)** Yaw is estimated by subtracting the relative yaw of the matched ground column from the absolute angle between the BEV pixel and each 2D pose. **(c)** BEV pixel (Orange) with relative yaw from matched column votes for a specific yaw for each pixel. Same column matches from multiple pixels (Yellow) along a line, e.g., a road, result in the same yaw being voted for pixels along the line (Red), while voting different yaws outside of the line (Grey).

tracts column-wise features from ground images and aligns them with dominant linear structures in BEV, such as roads, or multiple pixels in line from part of any structure. By aggregating alignment evidence via pairwise yaw voting, yaw estimation is reduced to a constrained 1-DoF problem. This formulation handles location ambiguity without requiring ground-view height or distance for explicit mapping, which otherwise introduces radial location uncertainty. This flexible decomposition reduces ambiguity and eliminates the need for assumptions about ground height.

This formulation has two broader implications. First, by decoupling yaw from translation, we reduce the complexity of the search space, enabling localization systems to operate more robustly. Second, precise yaw estimation serves as a reusable module that can enhance downstream localization pipelines, improving full 3-DoF methods beyond what end-to-end optimization alone can achieve. Experiments on Mapillary Geo-Localization (Sarlin et al., 2023a), Ford Multi-AV (Agarwal et al., 2020), KITTI (Geiger et al., 2013) and VIGOR (Zhu et al., 2021) confirm the effectiveness of LAYS, consistently outperforming state-of-the-art baselines, e.g., achieves 72.10% and 33.47% sub-degree accuracy on the MGL dataset with $45°$ and $180°$ noise setup, compared to 32.67% and 6.55% of the state-of-the-art. We further demonstrate that isolating the yaw estimation and explicitly reducing output to a 2-DoF space enables more precise localization for 3-DoF methods. This sets a new benchmark for precise yaw estimation and confirms a robust foundation for downstream localization pipelines.

**Contributions.** This paper makes the following contributions:

- We **formulate** LAYS, a new approach to cross-view yaw estimation that achieves degree-level precision without relying on ground-height assumptions.
- We **introduce** a line alignment framework that integrates column-wise feature, ground-BEV matching, and pairwise yaw voting, decoupling yaw from location uncertainty.
- We **demonstrate** consistent state-of-the-art improvements on Mapillary, Ford, and VIGOR, establishing yaw as a tractable and foundational subproblem in global pose estimation, and this isolation results in a further boost in the localization performance.

## 2 RELATED WORKS

**Vision-based Global Positioning System** In early days (Brejcha & Čadík, 2017), methods localized the image coarsely with scale (Weyand et al., 2016; Hays & Efros, 2008). Some localize orientation and position using ground feature databases (Yang et al., 2022) or city-scale Structure from Motion (Agarwal et al., 2009; Li et al.) with SLAM (Middelberg et al., 2014). Google Maps offers Visual Positioning System (Rajpurohit et al., 2024). These provide accurate localization but require a dedicated, costly, large-scale feature database on the target location. Sarlin et al. (2023a) and Wu et al. (2024) use publicly available 2D maps as input, while Sarlin et al. (2023b) employs multi-modal learning to construct a semantic map for localization. 2D maps provide rich semantics, such as roads and buildings, but they lack detailed visual cues for localization.

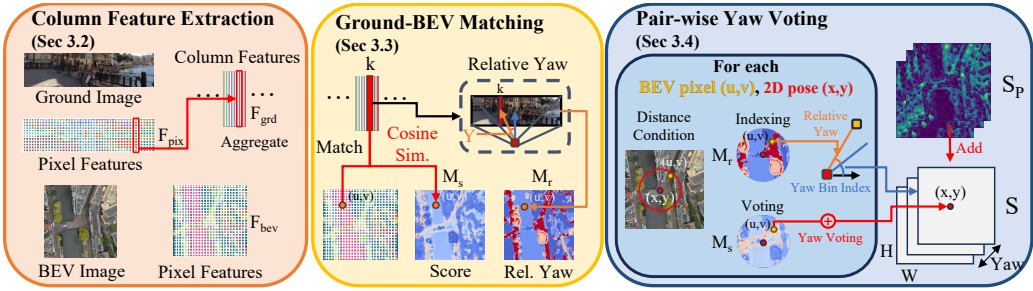

Figure 2: Overall framework, yaw-axis index of the maximum score bin is used as an estimated yaw.

**Visual Camera Rotation Estimation** Many rotation-focused pose estimation methods are developed for applications such as Unmanned Aerial Vehicle operations (Liu et al., 2022), where orientation is a critical input. Rotation estimation is often part of 6-DoF pose estimation, as location and rotation are typically inseparable in localization. Some studies focus on frame-to-frame rotation using omnidirectional cameras (Du et al., 2024) or handheld cameras in crowded scenes (Delattre et al., 2023). For 2D rotation, geometry-aware methods predict the horizon line from a single image (Xian et al., 2019), and roll and pitch estimation using camera parameters has been proposed (Jin et al., 2023). Unlike yaw, local features can estimate roll and pitch by using surrounding structural cues to align images with the direction of gravity. In a cross-view setup, rotation estimation is primarily in 2D by using gravity-aligned images.

**Ground and Aerial Cross-View Localization** Cross-view localization determines the global pose by matching a ground-level query image to geo-referenced satellite or aerial view data. Dedicated approaches bridge the significant modality and viewpoint gap between a perspective ground view and a top-down view. Early cross-view localization methods identified locations by comparing features from ground and satellite views from sparse pose candidates (Shi et al., 2019a; Liu & Li, 2019; Shi et al., 2019b; Guo et al., 2022; Shi et al., 2022; Zhu et al., 2022). This approach evolved to include orientation estimation (Shi et al., 2020) and matching of non-aligned images (Zhu et al., 2021). Recent methods estimate precise positions (Xia et al., 2022) and yaw in BEV by matching deep features (Shi & Li, 2022). These methods often assume a perspective transform with objects at a specific ground height (Fervers et al., 2023; Song et al., 2023; Wang et al., 2023a; 2024). Techniques include horizontally splitting images to match BEV features (Lentsch et al., 2023), using rolling descriptors for pose estimation (Xia et al., 2024), and refining homography estimates between BEV and projected ground images (Song et al., 2023) or using homography with panoramic images (Wang et al., 2023b). Wang et al. (2023a; 2024) iteratively refine the pose from an initial pose. Finally, Xia & Alahi (2025) maps ground view features to a 3D grid, finds a match between the 3D grid and BEV, and uses Procrustes analysis to estimate pose. Shi & Li (2022); Shi et al. (2023; 2024) uses yaw from simple joint regression or optimization of 3-DoF pose, focusing on slight yaw noise in the driving scene where the front of the ground image is aligned with the road. We propose an isolated yaw estimate applicable in more scenarios in the presence of extreme noise.

## 3 METHOD

### 3.1 OVERVIEW

LAYS framework is illustrated in Fig. 2. We follow standard cross-view localization input. Our approach takes as input an undistorted ground-level image $I_{grd} \in \mathbb{R}^{3 \times H_g \times W_g}$, approximately gravity-aligned along the $y$ axis, and a BEV image $I_{bev} \in \mathbb{R}^{3 \times H_b \times W_b}$, a top-down view that spatially covers the surrounding area. We assume that the camera intrinsics are given. Importantly, our formulation does not assume ground-truth location or ground height; the 2D pose $(x, y)$ remains fully unknown and is exhaustively searched during inference. The goal is to estimate yaw, formulated to output a 2D position-conditioned discrete yaw score bins $S(\theta, (x, y))$, a 3D array $S \in \mathbb{R}^{|\Theta| \times |\mathcal{X}| \times |\mathcal{Y}|}$, where $(x, y)$ represents the discrete 2D position in BEV coordinates, and $\theta$ denotes the yaw. The yaw from the 3D bin with the highest score across all 2D pixels and yaws is the estimated yaw.

Our relative yaw-based formulation is visualized in Fig. 1, where yaw is estimated by aligning lines in two views. BEV pixels belong to a BEV radial line that corresponds to a ground column. We match each BEV pixel with the ground column, which has a relative yaw from the camera front (Fig. 1-a). For each 2D pose, yaw can be estimated from an absolute angle between the 2D pose and the matched BEV pixel and the relative yaw of the matched ground column (Fig. 1-b). By voting match scores to the computed yaw, yaw can be estimated from the consensus of multiple confident matches on a BEV radial line (Fig. 1-c) despite ambiguity of location along the line. Matches from multiple pixels in lines of remarkable structures, such as roads or buildings, find yaw.

Three key components enable precise yaw estimation: **Column Feature Extraction** (Sec. 3.2) extracts column-wise features from the ground image that correspond to unique relative yaw. **Ground-BEV Matching** (Sec. 3.3) computes feature similarity scores between ground column and BEV pixel and selects one representative matching ground column for each BEV pixel. Finally, **Pairwise Yaw Voting** (Sec. 3.4) votes match scores for the estimated yaw for each pair of BEV pixel match and 2D candidate pose to find the most probable yaw. More details are in the appendix.

## 3.2 COLUMN FEATURE EXTRACTION

We extract features from the BEV and ground images for cross-view matching. The ground pixel features are aggregated column-wise, capturing features with unique relative yaw. This relative yaw is later used to estimate yaw for a pair of BEV pixel match and 2D pose. Relative yaw is not impacted by ground height or distance, relaxing the flat ground assumption.

**Pixel Feature Extraction** We employ two U-Net (Ronneberger et al., 2015) with a VGG-16 (Liu & Deng, 2015) encoder to extract $C$-dimensional features for two images following prior works (Shi et al., 2023; Wang et al., 2024). The BEV feature map is denoted as $\mathbf{F}_{\text{bev}} \in \mathbb{R}^{C \times H_b \times W_b}$. The ground feature map is extracted at the pixel level $\mathbf{F}_{\text{pix}} \in \mathbb{R}^{C \times H_g \times W_g}$. We add a 4th channel to the ground image before encoding, the heading embedding similar to Wang et al. (2023a), representing the relative yaw of each pixel.

**Column-wise Aggregation** We aggregate features column-wise to obtain a compact representation. For each column $k$, we weigh and sum the pixel features to extract highly relevant information. To compute the weight for pixel features, we use a fully connected layer $N$ and apply softmax. Given the feature of pixel $(k, j)$ as $\mathbf{F}_{\text{pix}}(k, j)$:

$$\mathbf{F}_{\text{grd}}(k) = \sum_{j=1}^{H_g} \frac{\exp\left(N(\mathbf{F}_{\text{pix}}(k, j))\right)}{\sum_{j'=1}^{H_g} \exp\left(N(\mathbf{F}_{\text{pix}}(k, j'))\right)} \mathbf{F}_{\text{pix}}(k, j) \tag{1}$$

This results in a column-wise feature $\mathbf{F}_{\text{grd}} \in \mathbb{R}^{C \times W_g}$.

**Feature Confidence Estimation** We compute confidence scores for both BEV and ground features to focus on distinct and matchable features. MLP attached to U-Net computes confidence from $\mathbf{F}_{\text{bev}}$ and $\mathbf{F}_{\text{pix}}$, producing $\mathbf{C}_{\text{bev}} \in \mathbb{R}^{H_b \times W_b}$ and $\mathbf{C}_{\text{pix}} \in \mathbb{R}^{H_g \times W_g}$. For each column, the highest pixel confidence is selected as $\mathbf{C}_{\text{grd}} \in \mathbb{R}^{W_g}$. These confidence scores are later used in the matching process to improve robustness by assigning higher importance to confident features.

## 3.3 GROUND-BEV MATCHING

We compute the cosine similarity scores of each pair of ground column and BEV pixel. One representative ground column is selected for each BEV pixel for an effective match. This process does not use projection. Instead, we keep the relative yaw from the ground column with a match score for each BEV pixel. This is used for voting yaw in Sec . 3.4 for each BEV pixel match and 2D pose.

**Match Score Computation** We use absolute cosine similarity between the BEV feature at pixel $(u, v)$ and each column feature of the ground image as a match score. Given BEV feature $\mathbf{F}_{\text{bev}}(u, v)$ and $k$-th ground column feature $\mathbf{F}_{\text{grd}}(k)$, the pairwise match score is:

$$\mathbf{P}((u, v), k) = |\mathbf{F}_{\text{bev}}(u, v) \cdot \mathbf{F}_{\text{grd}}(k)| \cdot \mathbf{C}_{\text{grd}}(k) \tag{2}$$

**Probabilistic Feature Selection**  To ensure that the most relevant column feature is selected for the match score, a column with the highest match score is selected for each BEV pixel for inference. To avoid overfitting to dominant features and avoid failure cases where no feature voting for ground-truth yaw is selected, we introduce a random variable to choose one of $W_g$ columns with probability proportional to the match score. Each matching BEV pixel $(u, v)$ chooses column $\mathbf{c}(u, v)$ as:

$$\mathbf{c}(u, v) = \begin{cases} \mathbf{c}_r, \text{ where } P(\mathbf{c}_r = k) = \frac{\mathbf{P}((u,v),k)}{\sum_j \mathbf{P}((u,v),j)}, & \text{if Training,} \\ \underset{k}{\arg\max} \, \mathbf{P}((u,v),k), & \text{otherwise} \end{cases} \quad (3)$$

Treating a feature with the opposite sign as equivalent to ensure the match score is positive, the final BEV pixel match score is computed as:

$$\mathbf{M_s}(u, v) = |\mathbf{C}_{\text{bev}}(u, v) \cdot \mathbf{P}((u, v), \mathbf{c}(u, v))| \quad (4)$$

**Relative Yaw Mapping**  Each ground column $k$ corresponds to a unique relative yaw $\mathbf{Y}(k)$, computed from the 3D ray direction in the camera's coordinate frame. We map the selected match to the relative yaw:

$$\mathbf{M_r}(u, v) = \mathbf{Y}(\mathbf{c}(u, v)). \quad (5)$$

After this process, each BEV pixel $(u, v)$ has an associated match score $\mathbf{M_s}(u, v)$ and a relative yaw $\mathbf{M_r}(u, v)$. These values are utilized in the Yaw Voting to estimate yaw.

### 3.4 Pair-wise Yaw Voting

Yaw estimation is ambiguous without a known 2D pose $(x, y)$. To resolve this ambiguity, we construct a 3D score bin $S(\theta, (x, y))$ over the full pose search space. We iterate over all candidate 2D poses. For each candidate $(x, y)$, we compute the conditional yaw that aligns each matched BEV pixel $(u, v)$ with the radial line of the ground column, and cast its match score $M_s(u, v)$ into the corresponding yaw bin. This process isolates yaw through a relative yaw-based formulation (Fig. 1-(b)), where correct BEV pixel matches along a single linear structure agree on the same yaw for all $(x, y)$ candidates on the line (Fig. 1-(c)). This line consensus removes the need for height-assuming projection and enables sub-degree yaw estimation under high location uncertainty.

**Yaw Estimation**  We compute the absolute angle $\theta_{\text{abs}}$ in BEV coordinate for each candidate pose $(x, y)$ relative to a matched BEV pixel $(u, v)$. The absolute yaw is given by:

$$\theta_{\text{abs}}((x, y), (u, v)) = \arctan\left(\frac{v - y}{u - x}\right) \quad (6)$$

The estimated yaw is then computed by subtracting the matched ground-view relative yaw $\mathbf{M_r}(u, v)$ from the absolute angle, as visualized in Fig. 1:

$$\theta_{\text{est}}((x, y), (u, v)) = \theta_{\text{abs}}((x, y), (u, v)) - \mathbf{M_r}(u, v) \quad (7)$$

**Yaw Voting**  We prepare a 3D score bin representation $\mathbf{S} \in \mathbb{R}^{|\Theta| \times (|\mathcal{X}| \times |\mathcal{Y}|)}$, where each bin corresponds to a yaw with pose condition $(\theta, (x, y))$ and $\mathcal{X}, \mathcal{Y}, \Theta$ are valid range of $x, y, \theta$ pose. The 2D poses are discrete per pixel, and yaw bins are defined per degree, both within the noise range specified by the dataset. The match score $\mathbf{M_s}(u, v)$ is accumulated to the corresponding yaw bin of $\theta$ for each candidate pose $(x, y)$, if $(u, v)$ is within distance $R$ from $(x, y)$:

$$\mathbf{S_M}(\theta, (x, y)) = \sum_{u,v} \mathbf{M_s}(u, v) \cdot \mathbb{I}\Big(\theta = [\theta_{\text{est}}((x, y), (u, v))] \wedge d((x, y), (u, v)) \leq R\Big), \quad (8)$$

where $\mathbb{I}$ is an indicator function ensuring voting only for matching yaw bins, $d((x, y), (u, v)) = \sqrt{(x - u)^2 + (y - v)^2}$. The yaw selection operation is not differentiable; however, the update of $\mathbf{M_s}$ impacts the selected match and its score as well as the estimated yaw.

**Pose Plausibility**  Different poses have different likelihoods due to the presence of roads and structures. We add a pose plausibility score $\mathbf{S_P}(\theta, (x, y))$ derived from the BEV features with an MLP that gives a hint for the estimation.

$$\mathbf{S}(\theta, (x, y)) = \mathbf{S_P}(\theta, (x, y)) + \mathbf{S_M}(\theta, (x, y)) \quad (9)$$

**Multi Level Score**  We use a 3-level estimate, with different resolutions, similar to Shi et al. (2023). One lower level has half the resolution, with a halved number of angular bins. Each level uses a per-pixel 2D pose, and the highest level uses a 1-degree angle bin. Lower-resolution scores are interpolated and added to higher-resolution scores. The highest resolution is used for the final estimate.

Applying softmax over the final pose candidate scores for each resolution, we compute the loss term to maximize the log probability of the ground-truth pose $(\theta, (x, y))$ to encourage high match scores to the correct line-alignment.

## 4 EXPERIMENTS

### 4.1 EXPERIMENT SETUP AND METRICS

To evaluate the method under representative real-world conditions where GPS and sensor data can be noisy, we add perturbations to the ground truth poses in both position and orientation, following the protocols of prior works (Xia et al., 2024; Shi et al., 2023). We focus on assessing the angular accuracy for yaw. The angular accuracy is measured using the ratio of the correct yaw estimate, where the estimation is correct if the yaw angle error is less than $\theta$ degrees relative to the ground truth. Consistent with previous studies (Wang et al., 2023a; 2024), we use thresholds of $\theta = 1°$, $2°$, and $4°$.

### 4.2 BASELINES

We compare LAYS against four state-of-the-art baseline approaches with code available: CCVPE (Xia et al., 2024), BoostAcc (Shi et al., 2023), G2S (Shi et al., 2024), and FG2 (Xia & Alahi, 2025). CCVPE utilizes rolling descriptors with orientation-map-based pose estimation. BoostAcc and G2S utilize an optimizer and Spatial Transformer Networks to estimate 3-DoF poses, taking only yaw to compute correlations for 2-DoF poses. G2S uses self-supervised learning with only BEV. FG2 estimates a 3-DoF pose jointly by weighted matching of 3D grid-mapped ground view and BEV, utilizing Procrustes analysis.

We use the same training and test splits for the baseline methods and ours. We use official codes to train baselines on the Ford and MGL datasets, as they did not evaluate them on the dataset with our noise setup. On the VIGOR dataset, we use open-sourced weights for evaluation results, unless noted as re-trained due to unavailability.

### 4.3 DATASET

We evaluate on three datasets: (1) Mapillary Geo-localization dataset, a crowd-sourced dataset providing a greater variety of images not only on the road, which suits AR/MR scenarios, (2) Ford Multi-AV Seasonal dataset, which represents a highway driving scene where minor orientation errors can be impactful. (3) VIGOR dataset, featuring $360°$ panoramic ground view with comprehensive information, commonly used by recent cross-view localization methods.

**MGL (Mapillary Geo-Localization) Dataset** (Sarlin et al., 2023a)[1] is a crowd-sourced dataset comprising a variety of ground-images captured from vehicles, handheld cameras, and bicycles, rich with images not front-aligned with the road direction. The variety introduces challenge and helps evaluate performance in AR/MR applications where visual alignment is crucial. The dataset has precise SfM-refined ground truth with a ground image rectified to be gravity-aligned. We augment the Amsterdam split with Google Map, as it contains a variety of surroundings and has constant resolution applicable to cross-view localization approaches. Splitting by sequence, there are a total of 27,159 and 9,103 frames for the training and test sets with ground view resolution of $512 \times 512$.

**Ford Multi-AV Dataset** (Agarwal et al., 2020) evaluates driving scenes with the satellite view image from Google Map (Shi et al., 2023). Among the 6 logs from different areas, we use log 1 on the highway driving scene, where a small angular error is critical. We follow Shi et al. (2023) splits, ground images are resized to $256 \times 1024$, and satellite images are cropped to $512 \times 512$ with the noisy position at the center. Up to $\pm 20$m of lateral and longitudinal noise and $\pm 45°$ yaw noise is injected

---

[1]All images are publicly available under a CC-BY-SA license via the Mapillary API (Sarlin et al., 2023a).

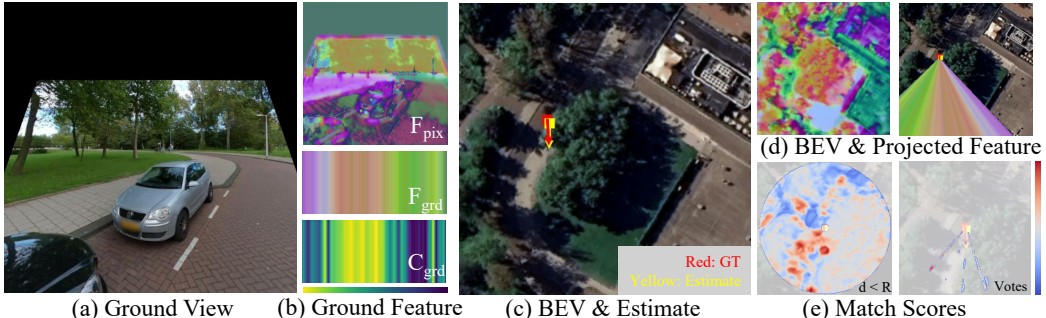

(a) Ground View     (b) Ground Feature     (c) BEV & Estimate     (e) Match Scores

Figure 3: Visualization of our yaw estimation in the MGL dataset. To visualize features, we used Principal Component Analysis (PCA). **(a)** is the ground view. **(b)** are the ground pixel, column features, and confidence: $\mathbf{F}_{pix}$, $\mathbf{F}_{grd}$, and $\mathbf{C}_{grd}$, with a colormap for confidence. **(c)** is the BEV image with the estimation result. **(d)** are the BEV and the projected ground feature for the estimated pose. The projection is solely for visualization purposes; it is not utilized by our method. **(e)** are match scores $M_s$ within an $R$ distance of the estimated pose and ones that vote for the estimated yaw, with a color map. Yaw is estimated by finding line correspondence of the road, despite differences in time and curvature. **More examples from all datasets are in the appendix.**

Table 1: Comparison of cross-view yaw estimation accuracy on MGL Dataset with $\pm20$m location noise under $\pm45°$ and $\pm180°$ yaw noise.

| Method | $\pm45°$ | | | $\pm180°$ | | |
|---|---|---|---|---|---|---|
| | $< 1°$ | $< 2°$ | $< 4°$ | $< 1°$ | $< 2°$ | $< 4°$ |
| CCVPE (Xia et al., 2024) | 21.36 | 40.78 | 68.15 | 6.55 | 12.55 | 23.79 |
| BoostAcc (Shi et al., 2023) | 7.40 | 14.19 | 27.54 | 0.58 | 1.09 | 2.13 |
| G2S (Shi et al., 2024) | 32.67 | 57.49 | 82.50 | 0.59 | 1.27 | 1.86 |
| FG2 (Xia & Alahi, 2025) | 18.59 | 34.97 | 59.20 | 3.13 | 6.28 | 12.44 |
| **Ours** | **72.10** | **90.16** | **95.63** | **34.81** | **52.42** | **61.74** |

into the ground truth pose, following (Wang et al., 2024). The dataset has fixed camera calibration matrices.

**KITTI Dataset** (Geiger et al., 2013) is widely used dataset for cross-view localization benchmark. The KITTI dataset has one training set and two test sets, one from the same areas as the training set and the other from different areas. We follow Shi et al. (2023) for splits and have ground- and satellite-view images resized to the same size as in the Ford dataset. Up to $\pm20$m of lateral and longitudinal noise, and $\pm180°$ of yaw noise, are injected into the ground truth pose for our evaluation.

**VIGOR Dataset** (Zhu et al., 2021) is a dataset commonly used by recent cross-view localization methods, containing ground panorama images of four cities. We follow standard protocols using only positive samples and locational noise range (center half width and height of the BEV image) for fine-grained cross-view localization (Lentsch et al., 2023; Xia et al., 2024). There are two splits: Same area has all cities as training and test datasets, and cross area has two cities as the training set and the other two as the test set. We use the unknown orientation setup with $\pm180°$ angular uncertainty for yaw estimation.

### 4.3.1 RESULTS

Table 1 presents the cross-view angular localization accuracy of LAYS compared to baseline approaches on the **MGL** dataset. This dataset presents significant challenges due to the complexity of urban environments and the increased variability, which introduces ambiguity to the height that projection-based methods depend on. Consequently, these methods struggle to estimate angles reliably in this scenario. In contrast, LAYS achieves a sub-degree accuracy of 72.10%, over doubling the performance of the next best method G2S, which attains 32.67%. These results underscore the robustness of our approach in challenging real-world conditions.

Table 2: Comparison of cross-view yaw estimation accuracy on Ford Dataset Log1 (Highway) with $\pm20$m location noise and $\pm45°$ yaw noise.

| Method | $<1°$ | $<2°$ | $<4°$ |
|---|---|---|---|
| CCVPE (Xia et al., 2024) | 41.36 | 62.74 | 75.11 |
| BoostAcc (Shi et al., 2023) | 24.38 | 43.24 | 67.86 |
| G2S (Shi et al., 2024) | 16.81 | 47.71 | 68.43 |
| FG2 (Xia & Alahi, 2025) | 16.75 | 41.84 | 57.13 |
| **Ours** | **67.05** | **85.14** | **91.76** |

Table 3: Comparison of cross-view yaw estimation accuracy on KITTI dataset with $\pm20$m location noise and $\pm180°$ yaw noise. *: CCVPE provides results with unknown orientation on KITTI, with $<1°$, $<3°$, and $<5°$ accuracy, we take those values to show upperbound for the $<2°$ and $<4°$ degree accuracy.

| Method | Test 1 (Same Area) | | | Test 2 (Cross Area) | | |
|---|---|---|---|---|---|---|
| | $<1°$ | $<2°$ | $<4°$ | $<1°$ | $<2°$ | $<4°$ |
| CCVPE* (Xia et al., 2024) | 8.96 | <26.48 | <42.75 | 3.14 | <9.24 | <14.56 |
| BoostAcc (Shi et al., 2023) | 0.53 | 0.90 | 1.96 | 0.56 | 1.05 | 2.09 |
| G2S (Shi et al., 2024) | 0.58 | 1.17 | 2.17 | 0.54 | 0.82 | 1.64 |
| FG2 (Xia & Alahi, 2025) | 1.62 | 3.15 | 6.47 | 1.62 | 2.96 | 6.13 |
| **Ours** | **48.69** | **58.65** | **60.43** | **45.51** | **53.34** | **54.44** |

$180°$ yaw noise is a particularly challenging setup, as methods are typically evaluated either with slight angular noise or with a panoramic image with $360°$ visibility (e.g., VIGOR). Unlike $45°$ noise, this challenging setup leads to failure of the existing method's rotation estimation. Our method achieves 33.47% sub-degree accuracy while the following best method, CCVPE, achieves 6.55%. Shi et al. (2023; 2024) using a simple 3-DoF model to estimate yaw struggle with high yaw uncertainty.

Table 2 shows the cross-view angular localization accuracy on the **Ford Multi-AV** dataset. LAYS significantly outperforms existing approaches, achieving improvements of 25.69% in subdegree angular localization accuracy. Our method is trained and tested without considering pitch and roll deviations due to noise, consistent with prior works. This demonstrates robustness against possible sensory noise during the drive despite the assumption of gravity-aligned images.

We evaluate performance on the **KITTI** dataset in Table 3. Most methods have already been evaluated on the KITTI dataset with a small noise range of up to $\pm10°$, where they perform well. With unknown yaw, however, most methods break down, similar to what has been seen in the MGL dataset with a limited FoV. Our method still achieves 48.69% and 45.51% sub-degree accuracy with an unknown yaw setup, while the second-best performing CCVPE achieves 8.96% and 3.14% accuracy.

Table 4: Comparison of cross-view yaw estimation accuracy on VIGOR Dataset with $\pm0.25$ BEV width and height location noise and $180°$ yaw noise. *: BoostAcc did not evaluate on VIGOR. **: G2S originally evaluated only the translation estimator on VIGOR. †: FG2 provides VIGOR result with a two-stage approach.

| Method | Same Area | | | Cross Area | | |
|---|---|---|---|---|---|---|
| | $<1°$ | $<2°$ | $<4°$ | $<1°$ | $<2°$ | $<4°$ |
| CCVPE (Xia et al., 2024) | 8.31 | 16.69 | 32.40 | 8.96 | 17.66 | 34.34 |
| BoostAcc (Shi et al., 2023)* | 0.50 | 1.10 | 2.16 | 0.64 | 1.20 | 2.30 |
| G2S (Shi et al., 2024)** | 2.33 | 4.60 | 8.92 | 3.98 | 7.72 | 14.45 |
| FG2 (Xia & Alahi, 2025) | 19.05 | 35.94 | 58.91 | 9.88 | 19.37 | 35.57 |
| FG2 (Xia & Alahi, 2025)† | 20.78 | 38.17 | **62.11** | 12.39 | 23.48 | 41.40 |
| **Ours** | **34.16** | **50.27** | 59.32 | **23.67** | **37.09** | **47.15** |

Table 5: Impact of components in LAYS for yaw estimation with $180°$ yaw ambiguity in MGL dataset.

| Alternative Method | $<1°$ | $<2°$ | $<4°$ |
|---|---|---|---|
| Point-to-Point Match | 13.19 | 23.67 | 38.88 |
| Keypoint-based | 20.12 | 28.75 | 33.10 |
| Column MLP | 31.73 | 48.23 | 58.13 |
| Greedy Selection | 28.69 | 41.70 | 47.62 |
| Test-time Weighted | 30.77 | 44.72 | 51.33 |
| **Ours** | **34.81** | **52.42** | **61.74** |

Table 6: Our yaw estimator's capability on improving the existing cross-view localization method (Xia & Alahi, 2025)'s accuracy, in the cross-area setup of the VIGOR dataset.

| Method | Mean Error↓ |
|---|---|
| FG2 | 10.02 |
| + Ground-truth Yaw | 2.41 |
| + Two Stage | 5.95 |
| **+ Ours Yaw** | 5.43 |
| **+ Ours Yaw** (2-DoF) | 5.10 |

The driving setup dataset with an omnidirectional camera, the **VIGOR** dataset result is demonstrated in Table 4. Despite significant angular noise, the panoramic view provides more information. We observed that our method performs best on this dataset, while the prior method does not break down entirely due to omnidirectional visuals. Shi et al. (2024) did not officially evaluate their rotation estimator in this setup, due to the difficulty in aligning the panoramic ground view with the satellite view when training only on the satellite view. Xia & Alahi (2025) demonstrates decent accuracy by leveraging omnidirectional features. They further employ a two-staged inference to mitigate high angular noise; however, adding more stages does not lead to a significant improvement in yaw accuracy.

### 4.4 ANALYSIS

#### 4.4.1 MATCHING PROCESS AND LOCATION

Fig. 3 and the examples in the appendix illustrate the matching process using our method. The red color indicates highly contributing BEV matches to the selected pose estimate. High-scoring matches arise along the road visuals, and these matches vote all locations along the road in the same lane for the same yaw (as in Fig. 1-c). With multiple remarkable visuals in some examples, our method estimates location; however, general results confirm that our approach primarily focuses on a specific line for yaw estimation, with location error along the primary matching line, both for the training and test sets. Our formulation does not concentrate scores on a specific pixel location. Scores are added to multiple negative samples from all pixels; as a result, our method learns to find only the most confident line correspondences. Instead, our method can support 3-DoF localization methods as shown in Sec. 4.4.3, reducing their search space to 2-DoF.

#### 4.4.2 ABLATION STUDY

We perform an ablation study to evaluate the impact of each component in LAYS, as summarized in Table 5. First, in **Point-to-Point Match**, we show the importance of line-based yaw scoring compared to projection-based baseline with 3-DoF scoring. This project and maps ground pixels to BEV for every possible 3-DoF pose candidate, using an FFT convolution operator from Sarlin et al. (2023a) to calculate match score $S_M$, and retaining the rest of our framework. While costly and exhaustive, 3-DoF scoring shows decent results, albeit significantly behind line-based scoring for yaw estimation.

We analyze Column Feature Extraction (Sec. 3.2). In **Keypoint-based**, we use a keypoint feature extractor employed by Wang et al. (2023a). This leads to poor performance, as the keypoint often fails to converge on important features. **Column MLP** uses MLP to aggregate pixel features for the final column feature. The weighted sum formulation has better accuracy overall.

Finally, we show the effectiveness of the train-time only weighted selection. In **Greedy Selection**, we match each BEV pixel to the ground feature with the highest correlation score. This choice leads to some dominant features overshadowing others, thereby limiting the diversity of information used. **Test-time Weighted** uses probabilistic matching during the test-time. While yielding decent results, we found that the highest-scoring match proved more effective in inference.

### 4.4.3 ACCURATE YAW'S IMPACT ON LOCALIZATION METHOD

We further explore the potential to enhance the 3-DoF localization method, and show the effectiveness of isolating the problem to the 2-DoF space. Granted, 3-DoF localization methods that separate yaw estimation from translation estimation, such as Shi & Li (2022); Shi et al. (2023; 2024), can be improved by replacing their rotation estimator. We demonstrate that a fully integrated 3-DoF estimation method can be improved by reducing its problem space, using our yaw estimator, using the example of Xia & Alahi (2025), that matches a 3D grid mapped from ground view and BEV, and uses Procrustes analysis for 3-DoF pose.

Table 6 compares the localization results of FG2 (Xia & Alahi, 2025) under different yaw configurations. We show their reported result, with unknown and known initial orientation. Their result is improved by using a two-stage inference for VIGOR with high yaw uncertainty, utilizing one model for a better initial yaw estimate and another to estimate the final 3-DoF pose. In **+ Ours Yaw**, we replace the first model with our more accurate model, and **+ Ours Yaw (2-DoF)** modifies FG2 to fixed yaw given by our method for Procrustes Analysis, estimating only the 2D location. Although both ways improve the accuracy, explicit reduction of the problem space leads to further improvement, 14%, indicating that isolation of rotation and location estimation has more potential to explore.

## 5 CONCLUSION

We introduced LAYS, a novel method that focuses on aligning lines, achieving sub-degree precision in yaw angle estimation through cross-view matching between the bird's-eye view and ground images. Prior methods require an exact location to find yaw and rely on assumptions about ground height. LAYS employs gravity-aligned ground column features compared with BEV pixels to find the corresponding radial line. A relative yaw-based formulation estimates yaw for each 2D pose and BEV match pair in location uncertainty. Multiple agreeing votes find the final yaw. LAYS achieves state-of-the-art performance in yaw estimation for both AR, MR, and driving scenarios. Not only a significant improvement in the yaw estimation, but the fundamental isolation of yaw estimation holds promise for cross-view localization with reduced dimensionality, advancing precise and robust cross-view localization systems.

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

APPENDIX

In this appendix, we provide the following:

- Training setup (Sec. A)
- Pseudo-code for Pair-wise Yaw Voting (Sec. B)
- Implementation Detail (Sec. C)
- Preprocess for Mapillary Geo-Localization Dataset (Sec. D)
- Robustness Against Pitch and Roll noise (Sec. F)
- Visualization of Matching Process (Sec. E)
- Discussion and Future Works (Sec. G)
- LLM Usage Disclaimer (Sec. H)

## A  TRAINING SETUP

For the training setup, we based our code on top of Shi et al. (2023)'s code. We use NVIDIA RTX A6000 for training, with batch size 6 and learning rate $10^{-4}$, using AdamW (Loshchilov & Hutter, 2019) optimizer. Loss weight $e^{-5}$ is used to adjust the loss scale. StepLR from PyTorch (Paszke et al., 2019) is used, with a gamma of $0.25$ and step size of $4000$ iterations.

## B  PSEUDO-CODE FOR PAIR-WISE YAW VOTING

---

**Algorithm 1** Pair-wise Yaw Voting with Distance Constraint

---

1: **Input:** Match yaw array $\mathbf{M_r}$, Match score array $\mathbf{M_s}$, Pose plausibility $\mathbf{S_P} \in \mathbb{R}^{|\Theta| \times (|\mathcal{X}| \times |\mathcal{Y}|)}$
2: **Output:** Yaw bins for 2D poses $\mathbf{S} \in \mathbb{R}^{|\Theta| \times (|\mathcal{X}| \times |\mathcal{Y}|)}$
3: $\mathbf{S} \leftarrow$ Zero-filled array $\in \mathbb{R}^{|\Theta| \times (|\mathcal{X}| \times |\mathcal{Y}|)}$
4: $\theta_{\max} \leftarrow$ Max yaw noise (degrees)
5: $R \leftarrow$ Max distance radius
6: **for all** each candidate pose $(x, y) \in \mathcal{X} \times \mathcal{Y}$ **do**
7:     **for all** each BEV pixel $(u, v)$ in All BEV pixels **do**
8:         $d \leftarrow \sqrt{(u - x)^2 + (v - y)^2}$             ▷ Calculate distance
9:         **if** $d < R$ **then**             ▷ Apply distance constraint
10:             $\theta_{\text{abs}} \leftarrow \arctan\left(\frac{v - y}{u - x}\right)$         ▷ Absolute angle
11:             $\theta_{\text{rel}} \leftarrow \mathbf{M_r}(u, v)$             ▷ Relative yaw
12:             $\theta_{\text{est}} \leftarrow \theta_{\text{abs}} - \theta_{\text{rel}}$            ▷ Estimated yaw
13:             $B_{\text{est}} \leftarrow \lfloor \theta_{\text{est}} + 0.5 \rfloor$           ▷ Yaw bin index
14:             **if** $-\theta_{\max} \leq B_{\text{est}} \leq \theta_{\max}$ **then**
15:                 $\mathbf{S}(B_{\text{est}}, (x, y)) \mathrel{+}= \mathbf{M_s}(u, v)$     ▷ Vote match score to yaw
16:             **end if**
17:         **end if**
18:     **end for**
19: **end for**
20: $\mathbf{S} \leftarrow \mathbf{S} + \mathbf{S_P}$                 ▷ Pose plausibility adjustment

---

We provide the pseudo-code of the Yaw-aligned Bin Score Scattering method's key component, score accumulation in Alg. 1, for clarification of its unique formulation. We wrote it in a loop form to make it more intuitive. PyTorch functions were used for effective implementation and GPU-accelerated parallelism. Further implementation details are in Sec . C.

The $\mathbf{S}$ array is indexed with coordinates and discrete in the pseudo-code. $\mathcal{X}$ and $\mathcal{Y}$ are all valid $x$ and $y$ 2D camera pose's coordinate in BEV space within noise range, and $\Theta$ is an valid discrete angles in degree $[-\theta_{max}, -\theta_{max} + 1, ..., \theta_{max} - 1, \theta_{max}]$, which depends on the maximum yaw noise $\theta_{max}$.

## C IMPLEMENTATION DETAIL

This section provides a detailed formulation of the method not discussed in the main paper for brevity. The variables denoted using the notation of a hat, such as $\hat{x}$ instead of $x$, are used for implementation.

### C.1 FEATURE EXTRACTION

We utilize the implementation of U-Net (Ronneberger et al., 2015) built on the VGG-16 (Liu & Deng, 2015) encoder backbone, which features separate weights for the ground and the BEV, as provided in the official code of Shi et al. (2023), along with an additional heading encoding channel. Instead of the cosine-based representation used by Wang et al. (2023a), we use linear mapping of the $x$ coordinate from $-1$ to $1$ and concatenate them along the channel axis.

Unlike the original implementation of Shi et al. (2023), the features are not normalized per image. Instead, ground and BEV features are normalized along channel dimensions:

$$\hat{\mathbf{F}}_{\text{bev}}(u, v) = \frac{\mathbf{F}_{\text{bev}}(u, v)}{\|\mathbf{F}_{\text{bev}}(u, v)\|_2}, \quad \hat{\mathbf{F}}_{\text{grd}}(u, v) = \frac{\mathbf{F}_{\text{grd}}(c)}{\|\mathbf{F}_{\text{grd}}(c)\|_2} \tag{10}$$

$\hat{\mathbf{F}}_{\text{bev}}$ and $\hat{\mathbf{F}}_{\text{grd}}$ are used for the cosine-similarity computation explained in the method section.

The confidence for each feature is computed using the original feature before normalization. For the ground confidence estimator, output values are directly from the dense layer, while the BEV confidence estimator values are the sigmoid of the dense layer output. Max normalization is applied to the BEV confidence value, so the maximum confidence is 1.

$$\mathbf{C}_{\text{bev}}^{\max} = \max_{u,v} \mathbf{C}_{\text{bev}}(u, v) \tag{11}$$

$$\hat{\mathbf{C}}_{\text{bev}}(u, v) = \frac{\mathbf{C}_{\text{bev}}(u, v)}{\mathbf{C}_{\text{bev}}^{\max}} \tag{12}$$

### C.2 PAIR-WISE YAW VOTING WITH DISTANCE CONSTRAINT

We only consider possible positions under the assumption of the noise range for the score bins. The score bin is $\mathbb{R}^{|\Theta| \times (|\mathcal{X}| \times |\mathcal{Y}|)}$. $\mathcal{X}$ and $\mathcal{Y}$ are the sets of ranges for the $x$ and $y$ of pixels covering the maximal location noise range:

$$\mathcal{X} = \{[i\Delta x, (i+1)\Delta x] \,|\, i \in \{-L, \ldots, L-1\}\} \tag{13}$$
$$\mathcal{Y} = \{[j\Delta y, (j+1)\Delta y] \,|\, j \in \{-L, \ldots, L-1\}\} \tag{14}$$

, where $\Delta x$ and $\Delta y$ correspond to the same meter per pixel value of the BEV feature. Denoting maximal location noise along one direction as $N_{\textbf{max}}$, $L$ is:

$$L = \left\lceil \frac{N_{\textbf{max}}}{\Delta x} \right\rceil \tag{15}$$

$\Theta$ is the range of $\theta$ in degrees, from ceil of $-\theta_{\textbf{max}}$ to $\theta_{\textbf{max}}$ divided by each resolution's bin's angular interval $\Delta\theta$ explained in the Sec. C.3.

$$\Theta_i = \{((i-1)\Delta\theta, i\Delta\theta), i \in \{-n, \ldots, -1, 0, 1, \ldots, n-1\}\} \tag{16}$$

, where

$$n = \left\lceil \frac{\theta_{\max}}{\Delta\theta} \right\rceil \tag{17}$$

We choose a fixed $R = 36.32$, subtracting 20m from the maximal distance from the center $256 \times 0.22$m, for pixel length and meters per pixel for the Mapillary Geo-localization dataset (Sarlin et al., 2023a) and the Ford dataset (Agarwal et al., 2020). For the VIGOR dataset (Zhu et al., 2021), we use a quarter of BEV's length of one side for $R$, as the meter per pixel varies across different cities.

### C.3 POSE PLAUSIBILITY SCORE MODEL

The pose plausibility score $\mathbf{S_P}$ is computed with a model based on Shi et al. (Shi et al., 2023)'s Uncertainty estimator implementation. BEV features from a separate VGG-16 U-Net are used for input to the model, as sharing features disturbs the training of the main match scores. The intermediate layer dimension is adjusted to fit the output channel and number of angular bins for each resolution. The LeakyReLU (Xu et al., 2015) replaces the final non-linearity with a negative slope 0.01.

### C.4 MULTI RESOLUTION SCORING

The final score computation is done for each resolution's feature. The ground and BEV image features with the same downsampling ratio from the U-Net are used for each resolution. Following Shi et al. (Shi et al., 2023), we use features downsampled 2, 4, and 8 times for scoring, and the score from the 2 times downsampling features is used for the final estimation. Each resolution is supervised using the log probability loss, where the lower-resolution feature is used for coarser estimation.

The score bin is built per pixel for different coarseness levels for each resolution. The angular bin has discrete intervals of 1, 2, and 4 degrees for each resolution, with larger intervals for lower resolutions. The score is repeated along each axis and summed to a higher resolution. The summation is done before applying kernels for the scattering-based match score $S_M$ and the 3D plausibility score $S_P$.

### C.5 LOSS TERM

Given the bin score tensor $S \in \mathbb{R}^{|\Theta| \times (|\mathcal{X}| \times |\mathcal{Y}|)}$, the loss maximizes the score of the ground truth pose bin, inspired by a 2D map-based localization method (Sarlin et al., 2023a). For the ground-truth poses $x_{\text{gt}}$, $y_{\text{gt}}$, and azimuth angle $\theta_{\text{gt}}$, the loss maximizes the log probability of the ground-truth bin after computing probability with softmax for all bins. The formulation is:

$$\mathcal{L} = -log\left(\frac{e^{S(x_{\text{gt}}, y_{\text{gt}}, \theta_{\text{gt}})}}{\sum_{x \in \mathcal{X}} \sum_{y \in \mathcal{Y}} \sum_{\theta \in \Theta} e^{S(x,y,\theta)}}\right) \tag{18}$$

where $S$ is the score in the bin for given $x$, $y$, and $\theta$.

## D PREPROCESS FOR MAPILLARY GEO-LOCALIZATION DATASET

The dataset, paired with a 2D map, can be downloaded by following instructions from Orienter-Net (Sarlin et al., 2023a)'s repository. The data contains ground-view images with camera pose information. We use the urban Amsterdam part of the dataset for the cross-view setup. The Google static map API fetches satellite view images, and we filtered out top-down occluded views. Consistent with previous work (Shi et al., 2023; Wang et al., 2024), satellite views are obtained as $1280 \times 1280$ images with a zoom level of 18 and a scale of 2. The satellite view is fetched for each ground view; however, no new image is downloaded if one of the existing satellite images covers $512 \times 512$ around the ground view.

OrienterNet (Sarlin et al., 2023a), which proposes the Mapillary Geo-Localization Dataset, mentions that they removed ground views with low visibility of surroundings, such as those facing a wall. We found such images remained, and removed those proportions. The Amsterdam dataset is generally at $512 \times 512$ resolution; we filtered out a small number of images with different resolutions. The train and test sets are split by sequence index to minimize spatial overlap.

## E VISUALIZATION OF MATCHING PROCESS

In this section, we demonstrate the matching process and explain how our method accurately estimates yaw with more examples. We visualize the matching process using Principal Component Analysis (PCA). We extract three major components to represent it as an RGB. For each sample, ground column features and BEV features are sampled and re-scaled for PCA. Since we use an absolute similarity score, feature vectors are normalized by flipping the sign when they have a negative similarity with the mean of all feature vectors.

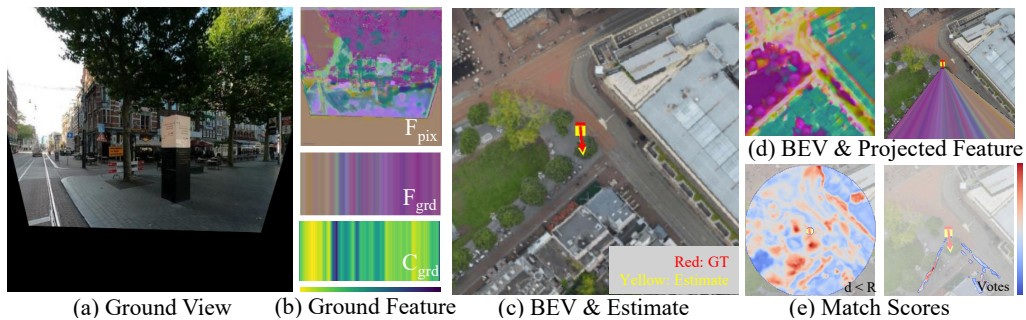

(a) Ground View     (b) Ground Feature     (c) BEV & Estimate     (e) Match Scores

Figure 4: Visualization of our yaw estimation in the MGL dataset. The estimated pose has the most votes from a tree.

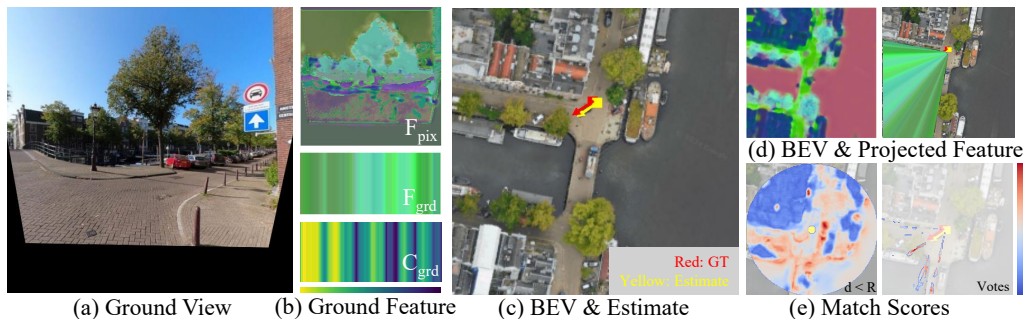

(a) Ground View     (b) Ground Feature     (c) BEV & Estimate     (e) Match Scores

Figure 5: Visualization of our yaw estimation in the MGL dataset. Our method gives high confidence and a match score for the column-aligned road on the left and the tree in the middle.

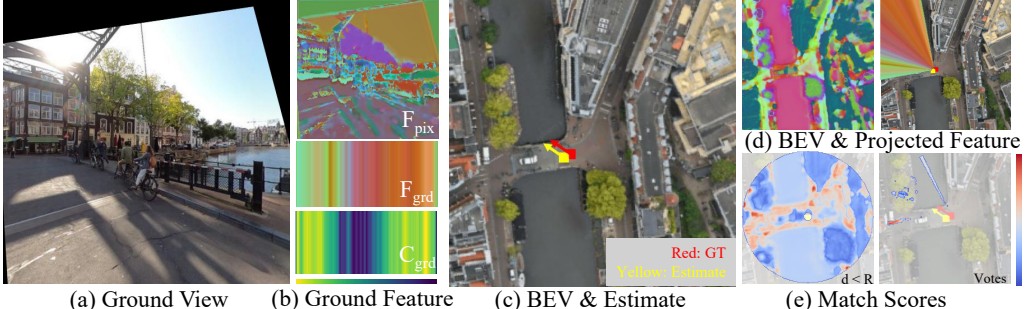

(a) Ground View     (b) Ground Feature     (c) BEV & Estimate     (e) Match Scores

Figure 6: Visualization of our yaw estimation in the MGL dataset. Our method has high confidence and a match score on the features extracted from the building at the front of the road.

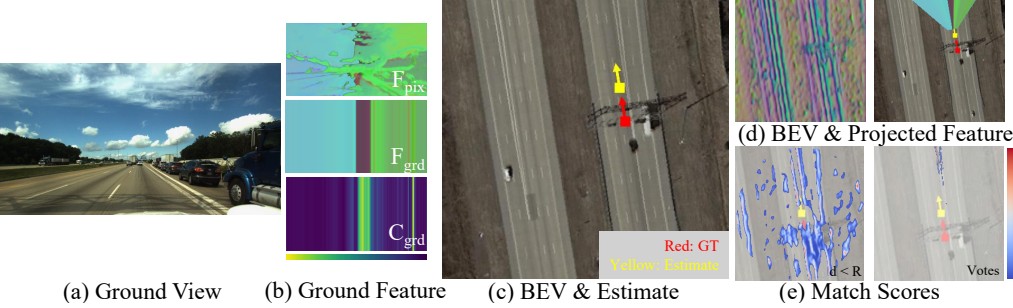

(a) Ground View     (b) Ground Feature     (c) BEV & Estimate     (e) Match Scores

Figure 7: Visualization of our yaw estimation in the Ford dataset. Our method aligns the road components.

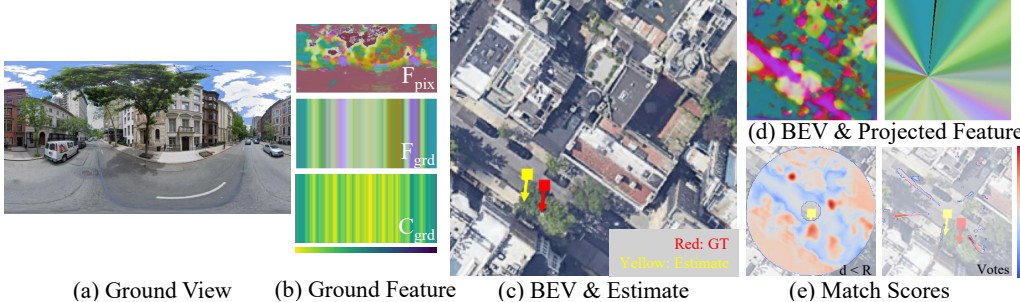

| (a) Ground View | (b) Ground Feature | (c) BEV & Estimate | (e) Match Scores |

Figure 8: Visualization of our yaw estimation in the VIGOR dataset, same-area setup. Our method aligns a specific building and road.

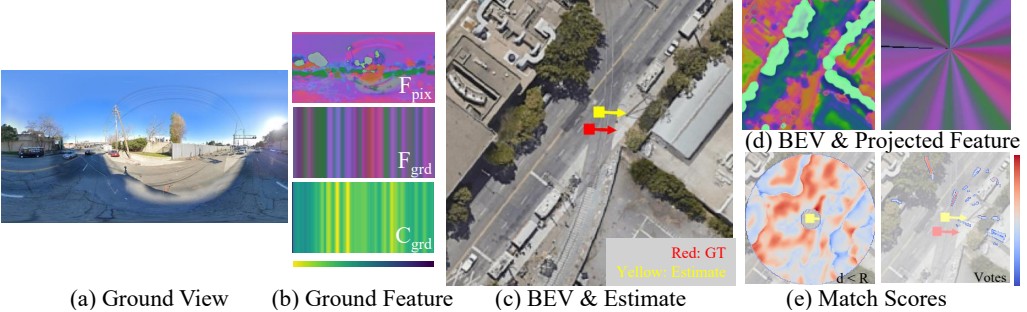

| (a) Ground View | (b) Ground Feature | (c) BEV & Estimate | (e) Match Scores |

Figure 9: Visualization of our yaw estimation in the VIGOR dataset, cross-area setup. Our method aligns the road and a tree.

Experiments in the MGL (Sarlin et al., 2023a) dataset revealed that our method not only focuses on the road, but also on remarkable ordinary road objects. Fig. 4 and Fig. 5 provide examples from the MGL dataset that focus on a tree, colored in dark purple and cyan, respectively. Fig. 6 shows the case where our process focuses on the building in the road front direction, with the column feature visualized as light brown on the left side of the ground image. Generally, our method estimates yaw based on the focused score in one direction.

The Ford (Agarwal et al., 2020) and VIGOR (Zhu et al., 2021) dataset results primarily focus on the driveway where the datasets were captured. An example from the Ford dataset is visualized in Fig. 7, with ground-view-related features resized to their original aspect ratio. The features from the left and right sides of the view appear different, due to positional encoding attached to the input image. Match scores are focused on road lanes, and the final estimation primarily comes from the lane in the front direction. Fig. 8 shows the matching in the VIGOR dataset, same-area setup. Our method identifies correspondences such as sidewalk boundary (brown) and driveway (purple). Fig. 9 illustrates the matching process using the VIGOR dataset, specifically the cross-area setup. At the road intersection, our method identifies correct road correspondence and finds yaw, focusing on a road (purple) and a tree (dark green) near the other road.

Table 7: Impact of pitch and roll noise on cross-view yaw estimation accuracy on Mapillary Geo-localization Dataset with 20m location noise and $180°$ yaw noise.

| Pitch, Roll Noise | $<1°$ | $<2°$ | $<4°$ |
|---|---|---|---|
| $±0°, ±0°$ | 34.81 | 52.42 | 61.74 |
| $±10°, ±0°$ | 34.36 | 50.26 | 58.61 |
| $±10°, ±10°$ | 33.15 | 48.98 | 58.60 |

## F    Robustness Against Pitch and Roll noise

Cross-view localization methods assume undistorted images, rectified along the gravity and horizon direction. Device-mounted cameras can be calibrated in advance to correct distortion. The camera's roll and pitch can be estimated from complementary sensors embedded in the device, such as accelerometers or IMUs (Kamal Mazhar et al., 2020), or using optical methods (Jin et al., 2023). In practical scenarios, pitch and roll estimated from a complementary system may not be precise. Our method is robust against such errors, shown by evaluation in the Ford dataset, where fixed camera parameters are assumed, and pitch and roll variation during the drive are ignored.

To further examine our method's robustness to such noise, we trained and tested it on the MGL dataset with pitch and roll noise. Following the experimental setup in the main paper, we add $\pm 20$m longitudinal and lateral location noise, $\pm 180°$ yaw noise, and, in the input image rectification process, additionally $\pm 10°$ roll and pitch noise. The result in Table 7 reveals that our method stays robust under such noise. For the roll noise, the image encoder can adapt to it, as it is visually estimable from the horizon, and our method does not rely on the correct pitch of the image since it extracts features per column.

## G    Discussion and Future Work

Our method has limitations for exact location estimation when used standalone, due to the formulation adding a match score to all pixels. A combination of line-aligning scoring with the conventional projection-based method has the potential for more accurate yaw and location measurements.

Further improving cross-view localization for outdoor AR and MR applications is a new direction; however, it poses a new challenge due to the significantly increased freedom of pose and the rich surrounding objects. Most existing vehicle-based works are built on the assumption that features are mapped to flat ground and roads in the vehicle dataset. While the MGL dataset provides a more varied environment, a less structured scene for AR and MR applications has more room for exploration.

The 2D map-based localization (Sarlin et al., 2023a; Wu et al., 2024) and the cross-view localization have different advantages. The 2D map provides semantic meanings of the surroundings, but does not offer exact visual cues. The BEV from the cross-view localization setup provides exact visual cues without semantic meaning. The fusion of two modalities would guide a new direction to more challenging AR and MR localization.

## H    LLM Usage

This section clarifies the use of LLMs in accordance with ICLR policy. LLM is used to find potential missing related works. The LLM also aided in polishing the writing, particularly in improving readability in the approach section and the syntax of formulations.

