# OpenReview forum: "Cross-View Yaw Estimation in Location Uncertainty with Line-Aligning Yaw Scoring"
_ICLR.cc/2026/Conference — Submitted to ICLR 2026_

### Official Review · Reviewer_6GEP · 2025-10-25

**Soundness:** 3
**Presentation:** 3
**Contribution:** 3
**Rating:** 6
**Confidence:** 3

**Summary:**

This paper tackles the challenging problem of accurate yaw estimation, which is critical for autonomous navigation and AR/MR applications but difficult due to the lack of direct geometric cues. The authors propose LAYS, a line-aligning yaw scoring method that reformulates yaw estimation as a one-degree-of-freedom problem independent of ground height or distance assumptions. Experimental results on several benchmarks, including Mapillary Geo-Localization, Ford Multi-AV, and VIGOR, demonstrate substantial improvements in sub-degree yaw accuracy over prior approaches.

**Strengths:**

1. The paper introduces LAYS, a line-aligning yaw scoring method that reformulates yaw estimation as a 1-DoF problem, which is both conceptually elegant and practically effective.

2. The idea of leveraging line correspondences between BEV and ground views provides strong geometric grounding and interpretability. The approach does not rely on assumptions about ground height or distance, improving robustness in real-world scenes.

3. The authors show that isolating yaw estimation as a 1-DoF (degree of freedom) problem can yield benefits in downstream full pose (e.g., 3-DoF) localization tasks, illustrating the practical value of decoupling yaw from translation estimation.
LAYS achieves significant improvements in sub-degree yaw accuracy across multiple benchmarks (Mapillary, Ford Multi-AV, VIGOR), clearly outperforming prior methods.

**Weaknesses:**

The analysis of failure cases appears limited: for instance, what happens when there are few dominant linear structures (roads, lanes) visible, or when the matching line correspondence is ambiguous?

**Questions:**

Could the authors clarify how the method handles cases where the dominant linear structure (e.g., road lane line) is absent, ambiguous, or heavily occluded (e.g., dense vegetation, parking lots)? How robust is LAYS in such scenarios?

Could the authors can utilize tools like GeoCalib [1] to handle inputs without gravity-alignment?

The method treats yaw estimation independently of translation (x,y) estimation, have the authors experimented with feedback loops where yaw estimation is used to refine translation?

[1]GeoCalib: Single-image Calibration with Geometric Optimization, ECCV 2024

---

> ### Author Response · Authors · 2025-11-21
>
> Dear Reviewer 6GEP, thank you for recognizing the potential of our core idea and method. We will answer your questions through this comment.
>
> ***Handling Lack of Linear Features and Failure Cases (W1, Q1)***:  Our method does not always require a clear linear structure. It often also estimates yaw from the agreement of multiple pixels on a non-linear object (e.g., a tree), as demonstrated in Figures 4, 5, and 9 in the appendix. Occlusion of linear structure is typical in the cross-view localization dataset, not only in parking lots but also on roads with vehicles. However, the existence of a common structure can be indirectly inferred from subtle visual cues and thus does not significantly affect the matching process.
>
> In failure cases, LAYS sometimes confuses similar-looking structures, estimates incorrect angles, fails to find confident correspondence, and yields a saturated score from multiple directions. We will add some visualization of such cases. LAYS still handles most cases, considering other visual structures (buildings, roadside trees). Our method has not been tested in a less-structured environment, and such a setup remains an unexplored area for the general cross-view localization problem, which is primarily explored in a road navigation setting.
>
> ***Handling input without gravity-alignment (Q2)***: Most cross-view localization datasets have a ground view that is already gravity-aligned. The MGL dataset consists of non-gravity-aligned images and uses SfM-based camera poses to rectify it in the dataloader. We experimented with additional roll and pitch noise (+/- 10 degrees) added for the rectification process to evaluate a real-world scenario in which input alignment may not be perfect, as in methods like GeoCalib. In the +-20m location noise and +-180 degree yaw noise setup, our method achieved 33.15%, 48.98% and 58.60% accuracy for the 1, 2, and 4-degree thresholds, compared to 34.81%, 52.42%, and 61.74% without roll and pitch noise. This result demonstrates that our method is robust against noise or error from complementary calibration methods. This is possible because the roll is relatively easy to estimate from visual using the horizon, which the image encoder can adapt to. Our column-wise, relative yaw-based formulation does not rely on height/depth information impacted by pitch error. We will discuss the robustness in the appendix.
>
> ***Use of yaw estimation to refine translation (Q3)***: Beyond one-way feedback, where our yaw estimation is used by the 3-DoF estimator as in our experiments, creating a feedback loop is a promising direction. We have not experimented with such a feedback loop idea, but a common 3-DoF pose estimator evaluates a cost or score for each pose candidate, and it can provide feedback to LAYS to search with a smaller candidate pose, while LAYS can give an even more accurate yaw. Our demonstrated ability to estimate yaw precisely can provide a robust foundation for such an iterative system, and we see this as a promising future extension.

---

> > ### Comment · Reviewer_6GEP · 2025-11-26
> >
> > Thank you for your detailed response. It has addressed my concerns, and I will maintain my original score.

---

### Official Review · Reviewer_N7WJ · 2025-10-30

**Soundness:** 3
**Presentation:** 3
**Contribution:** 3
**Rating:** 6
**Confidence:** 4

**Summary:**

This paper introduces LAYS, a novel approach for cross-view yaw estimation that decouples orientation estimation from location uncertainty. The key innovation lies in formulating yaw estimation as a line alignment problem between ground-level and bird's-eye view (BEV) images, rather than treating it as a byproduct of joint pose estimation. The method extracts column-wise features from ground images, matches them with BEV pixels, and employs a pairwise voting mechanism to estimate yaw without relying on ground height assumptions. Extensive experiments on three benchmarks demonstrate state-of-the-art performance, particularly under challenging noise conditions.

**Strengths:**

The paper makes several valuable contributions. The core idea of treating yaw estimation as an independent 1-DoF problem represents a significant shift from prevailing approaches that couple orientation with location estimation. The line alignment formulation is both novel and intuitive, effectively leveraging structural correspondences between perspectives.

The technical execution is thorough, with careful attention to feature extraction, matching, and voting mechanisms. The column-wise feature aggregation with relative yaw encoding is particularly clever, as it naturally handles viewpoint differences without explicit projection models.

Experimental validation is comprehensive, spanning multiple datasets with varying characteristics. The substantial improvements over current methods are convincing, especially the performance gains under ±180° yaw noise where existing approaches struggle. The ablation studies provide solid evidence for design choices, and the demonstration that LAYS can enhance existing 3-DoF methods by reducing their search space is practically significant.

**Weaknesses:**

While the method excels at yaw estimation, its standalone capability for precise location estimation appears limited. The formulation naturally distributes scores along radial lines, which is optimal for orientation but suboptimal for pinpointing exact positions. The paper briefly mentions this limitation but could more explicitly discuss the implications for practical deployment.

The computational requirements of the multi-resolution scoring and exhaustive matching aren't thoroughly analyzed. In applications like autonomous navigation or mobile AR, inference efficiency matters, and some discussion of computational trade-offs would strengthen the practical contributions.

The evaluation, while comprehensive across datasets, remains within relatively structured environments. The method's performance in highly unstructured scenes (e.g., natural environments without clear linear features) remains an open question, though this is perhaps beyond the paper's current scope.

**Questions:**

1. Given the method's strength in yaw estimation but limitations in precise localization, have you considered hybrid approaches that combine LAYS with complementary position estimation techniques? What would be the architectural implications?

2. Could you provide more insight into computational requirements and potential optimizations? For real-time applications, are there strategies to reduce the matching or voting complexity?

3. The method assumes gravity-aligned images. How sensitive is performance to residual pitch/roll errors that might occur in practical IMU-assisted systems?

4. The line alignment concept is powerful. Might this principle extend to estimating other parameters, such as camera height, by analyzing multiple line correspondences?

5. Some failure cases or challenging scenarios would be informative. Are there particular scene types or conditions where the line alignment assumption breaks down?

---

> ### Author Response · Authors · 2025-11-21
>
> Dear Reviewer N7WJ, thank you for your constructive feedback and recognition of our core ideas. We will explain the potential weaknesses and answer questions through this comment.
>
> ***Practical Use (W1) and Hybrid Approach (Q1)***: In addition to our main focus on yaw, our experiment in Section 4.4.3 and Table 5 demonstrates how LAYS can work together with a 3-DoF pose estimator to reduce its search space. Particularly, we tested a hybrid approach by feeding our LAYS-estimated yaw into the 3-DoF method FG2. The experiment revealed that reducing FG2’s pose space to 2D by fixing yaw to our estimate improved mean location error from 5.95 to 5.1.
>
> Our method can serve as an accurate rotation estimator, replacing dedicated rotation estimation architectures across multiple methods. BoostAcc (Shi et al., 2023) and G2S (Shi et al., 2024), use dedicated yaw estimators and search for a 2-DoF location with the estimated yaw. The demonstrated example method, FG2, runs the model twice on the VIGOR dataset with an unknown orientation setup, where the first model’s orientation output is used for the second inference to obtain the location.
>
> A more in-depth hybrid approach could be a future scope, such as combining our voting-based score with projective-mapping-based score (e.g. OrienterNet) for cross-view localization, or explicitly filtering out unlikely poses from the 3-DoF search space, reducing the cost of 3-DoF pose evaluation.
>
> ***Computational Requirements (W2, Q2)***: Multi-resolution scoring does not incur significant overhead, as the highest resolution part alone takes most of the computation. The exhaustive voting process incurs some overhead; however, other cross-view localization methods incur similar overhead from point- or pixel-wise comparisons between the ground view and BEV in large 3-DoF pose space. For example, the state-of-the-art localization method FG2 has 0.26s of latency for two-staged estimation, while our method has 0.16s of latency.
>
> Currently, the voting process is the computational bottleneck. In a practical scenario, many pixels can be excluded from the 2D pose based on context, e.g., a vehicle or a foot, and a simple 2D pose plausibility model for a filter. Furthermore, our implementation has room for improvement in efficiency because it relies on existing functions. It uses unfold to subtract the absolute angle from the matched relative yaw of BEV pixels, and uses the scatter-add function to index the corresponding yaw bin. This index computation process is similar to convolution, except that subtraction is used instead of multiplication; a better kernel implementation would significantly reduce the computational overhead.
>
> ***Performance on Unstructured Scenes (W3)***: Our method is not limited to linear structures such as roads. As our visualizations show, LAYS uses multi-pixel-on-a-line features, such as those from trees (Fig. 4, 5, 9).
>
> We recognize that performance can be limited in unstructured scenes, and this is a challenging, unexplored area for the cross-view localization task, which falls within the scope of future work.
>
> ***Robustness Against Pitch and Roll Noise (Q3)***: We experimented on the MGL dataset with unknown yaw, a +-20m location noise range, and random pitch and roll noise ranging from -10 to 10 degrees during image rectification during training and testing. We got 33.15%, 48.98%, and 58.60% accuracy at 1, 2, and 4-degree thresholds, respectively, compared to 34.81%, 52.42%, and 61.74% without noise. The image encoder can adapt to roll noise as it is estimable from horizon visual, and our formulation does not depend on the image’s pitch; thus, LAYS can be robust to some noise.
>
> ***Extension to Other Parameters (Q4)***: The principle from line alignment could be extended, though estimation in other dimensions, like camera height, would require different feature extraction methods or a different scoring process. The idea is an interesting direction for future work.
>
> ***Failure Cases (Q5)***: We observed that typical failure cases confuse similar-looking structures in the image, or fail to find a line of a particular feature to match, and have a saturated score from different directions. However, in most cases, some match scores from surrounding features (buildings, roadside trees) help distinguish them. We will visualize some failure case images in the appendix.

---

> > ### Comment · Reviewer_N7WJ · 2025-11-28
> >
> > Thank you for the detailed response. My questions have addressed. I will keep my positive score.

---

### Official Review · Reviewer_snYM · 2025-11-01

**Soundness:** 1
**Presentation:** 2
**Contribution:** 1
**Rating:** 0
**Confidence:** 4

**Summary:**

This paper estimates the Yaw rotation in cross-view localization problem. The core idea is to match BEV image pixels to ground-view image columns based on feature similarity. The paper presents results on certain datasets with sota performance.

**Strengths:**

1 The paper introduces a specific formulation for yaw estimation, framing it as a line alignment problem rather than a direct pixel-level correspondence task.

**Weaknesses:**

1 The work focuses solely on yaw estimation, assuming ground-truth position is given. This severely limits its practical utility for real-world applications like autonomous driving or VR/AR.

2 The methodological pipeline (Column Feature Extraction, Ground-BEV Matching, Pair-wise Yaw Voting) bears a strong resemblance to parts of frameworks like OrienterNet[1], which jointly solve for location and orientation. The paper fails to clearly articulate the fundamental novelty of its components beyond this existing work.

3 The experimental setup is limited. The absence of evaluations on standard benchmarks like KITTI raises concerns about the generalizability of the reported performance.

4 As shown in Table 3, the experimental performance is not good.

[1] OrienterNet: Visual Localization in 2D Public Maps with Neural Matching. CVPR 2023

**Questions:**

1 The experiments appear to be conducted under the assumption of a known, noise-free ground-truth position. Could you clarify this? In real-world scenarios, positional information (e.g., from GPS) is always noisy. How would your method perform with inaccurate position inputs, and what is the expected degradation in yaw estimation accuracy?

---

> ### Author Response · Authors · 2025-11-21
>
> Dear Reviewer snYM, we want to clarify that the multiple weaknesses stem from a significant misunderstanding.
>
> 1. ***We do not assume that the ground-truth position is known***. Instead, we explicitly solve and evaluate under significant location noise, following the standard setting in 3-DoF cross-view localization tasks.. Please refer to the ***General Response*** for a concise explanation of our location-uncertain setup and how LAYS handles it.
>
> 2. Our formulation is fundamentally different from OrienterNet. The concern appears to be based on an incorrect assumption about how LAYS estimates yaw under location uncertainty using relative yaw encoding. OrienterNet performs full 3-DoF estimation by computing ground-to-BEV correspondences via polar transforms and depth-plane mapping, which requires a known pose. However, LAYS uses pose-agnostic ground-to-BEV matching and estimates yaw backwards for each candidate 2-DoF pose using relative yaw. For LAYS, a single linear structure is sufficient to estimate yaw, and LAYS does not rely on depth or projective mapping.
>
> 3. We evaluate on Ford and VIGOR, which are widely used in cross-view localization, and additionally on MGL, which better reflects Mixed Reality scenarios where accurate rotation is crucial and camera orientations vary widely. KITTI largely resembles Ford/VIGOR in structure and contains very small yaw noise (±10°), where yaw estimation becomes trivial and existing methods already perform near-saturation. For completeness, we will train and test LAYS and baselines under our high-yaw-noise protocol on KITTI if required.
>
> 4. The concern appears tied to the incorrect assumption that we evaluate using the ground-truth location. In our location-uncertain setting, LAYS achieves substantially higher yaw accuracy than prior cross-view methods. LAYS is particularly effective on MGL and Ford, where the ground-view FoV is limited, a common real-world scenario, making point-wise BEV-ground matches unreliable.
>
> We believe this clarification addresses the incorrect assumption about our problem setup, which influenced several concerns. We hope this helps with a fair reassessment of our work.

---

> > ### Comment · Reviewer_snYM · 2025-11-27
> >
> > Thanks for your response.
> > 1. My concern regarding the assumption of known ground-truth positions has been addressed, and I will raise my score accordingly. However, since the paper also estimates the (x, y) position, could the authors clarify why this is not explicitly described as 3-DoF localization?
> > 2. The motivation behind Ground-BEV Matching and Pair-wise Yaw Voting is not clearly explained. Is Ground-BEV Matching intended for coarse yaw estimation? How does it work? Pair-wise Yaw Voting appears to perform fine-grained estimation of x, y, and yaw—this should be reflected in the caption of Figure 2.
> > 3. Evaluation on the KITTI dataset is necessary to validate the method's performance.
> > 4. While the method shows improvement, the results are still not entirely satisfactory. Could the authors comment on this limitation?

---

> ### Author Response · Authors · 2025-12-03
>
> Dear Reviewer snYM,
>
> We are glad to hear the primary concern regarding positional assumptions has been resolved. We address your follow-up questions below:
>
> **1. Why not explicitly describe it as 3-DoF localization?**
>
> While our method technically searches the full 3-DoF space ($x, y, \theta$), our “Line aligning” formulation isolates yaw estimation from location uncertainty by focusing on single-line correspondence.
> Matches between a ground column and a BEV line form a constraint along the BEV line with one agreed yaw. Our voting formulation aims to accumulate high scores for the correct yaw from this line correspondence.
> Positionally, the score is distributed evenly along these radial lines, thus leaving the location uncertainty. At the same time, the yaw votes from column-matched pixels on a line accumulate at the correct angle regardless of the depth ambiguity.  This allows us to recover a highly precise, "isolated" yaw estimate without the precise location constraint.
>
> **2. Motivation for Ground-BEV Matching & Pair-wise Yaw Voting**
>
> **Ground-BEV Matching (Sec 3.3)** is for **feature association** used to find evidence on which BEV pixel corresponds to which ground column, not coarse estimation. Pixel match alone is not meaningful due to the location uncertainty. It matches a BEV pixel to a ground-view column based on visual similarity, assigning a "Relative Yaw" (from the camera center) to that BEV pixel.
>
> **Pair-wise Yaw Voting (Sec 3.4)** is what makes the feature association meaningful, a core **consensus mechanism** that estimates yaw formulated to embrace location uncertainty along the radial line. It converts the "Relative Yaw" into an “Absolute Yaw” for every candidate pose $(x,y)$. By aggregating these votes, the correct yaw emerges from the consensus of multiple aligned features on a BEV line (the "line-alignment").
>
> **3. Evaluation on KITTI**
>
> We have now run LAYS on KITTI under our rigorous **high-yaw-noise protocol** (+-180 degree ambiguity, +-20m location ambiguity). In contrast, standard benchmarks often assume slight noise (+-10 degree, +-20m location ambiguity) acceptable for existing methods.
>
> Note that **we use the official code for prior methods built for the KITTI dataset**, the same as what we have done for other datasets whenever available. We alter only the yaw noise range to our setup, and attempt multiple different built-in hyperparameters for prior methods, including the default one, and report the best attempt.
>
> | Method | Test 1 (Same Area) $<1^{\circ}$ | Test 1 (Same Area) $<2^{\circ}$ | Test 1 (Same Area) $<4^{\circ}$ | Test 2 (Cross Area) $<1^{\circ}$ | Test 2 (Cross Area) $<2^{\circ}$ | Test 2 (Cross Area) $<4^{\circ}$ |
> | :--- | :---: | :---: | :---: | :---: | :---: | :---: |
> | CCVPE* | 8.96 | <26.48 | <42.75 | 3.14 | <9.24 | <14.56 |
> | BoostAcc | 0.53 | 0.90 | 1.96 | 0.56 | 1.05 | 2.09 |
> | G2S | 0.58 | 1.17 | 2.17 | 0.54 | 0.82 | 1.64 |
> | FG2 | 1.62 | 3.15  | 6.47 | 1.62 | 2.96 | 6.13 |
> | **Ours** | **48.69** | **58.65** | **60.43** | **45.51** | **53.34** | **54.44** |
>
> (\* CCVPE paper evaluates unknown yaw setup, which shows 1, 3, 5 degree accuracy, we use it to show an upper bound for 1, 2, 4 degree accuracy.)
>
> The trend is similar to the results on the MGL dataset with unknown yaw. LAYS achieved 48.69% and 45.51% sub-degree accuracy on the same-area and cross-area test sets, respectively, while the best-performing prior method, CCVPE, achieved <9% sub-degree accuracy. **LAYS achieves robust performance**, confirming that yaw estimation from a single line-alignment’s robustness extends to KITTI. Under this challenging setup, baseline methods fail dramatically despite training and evaluating with their official code and the KITTI implementation.
>
> **4. Performance Satisfaction**
>
> Our results represent a significant leap in performance for the most challenging, realistic scenarios. Particularly, LAYS succeeds in estimating yaw for **limited-FoV cameras with unknown orientation** (MGL, KITTI), a setting where prior methods fail almost completely. With the omnidirectional camera in VIGOR, LAYS achieves 23.67% accuracy (<1 degree), nearly doubling the performance of the next best method, 12.39% accuracy.
> This is significant enough to contribute to the 3-DoF location accuracy when used as a replacement for the yaw estimation stage (Section 4.4.3).
> In the context of cross-view localization under large uncertainty, doubling the SOTA accuracy (360-degree FoV camera, unknown yaw) and enabling localization in previously unsolvable settings (typical limited FoV, unknown yaw) are highly satisfactory advancements.

---

### Official Review · Reviewer_7Qns · 2025-11-10

**Soundness:** 2
**Presentation:** 2
**Contribution:** 2
**Rating:** 2
**Confidence:** 4

**Summary:**

This paper proposes LAYS (Line-Aligning Yaw Scoring), a novel 1-DoF yaw estimation framework that matches BEV pixels to ground-view columns via feature similarity and uses line correspondences for pairwise yaw voting, effectively decoupling yaw estimation from location uncertainty. It addresses the challenge of accurate yaw estimation in cross-view localization, where conventional methods struggle due to the lack of direct geometric cues and dependence on ground height assumptions. The proposed work achieves sub-degree yaw precision and substantial gains across multiple datasets (Mapillary, Ford Multi-AV, VIGOR), establishing yaw as an independent, solvable subproblem and improving global pose localization accuracy.

**Strengths:**

1. This paper presents a method that transforms a challenging 3-DoF rotation problem into a decoupled 1-DoF line-alignment task, enabling efficient and accurate yaw estimation without assuming ground height or distance.
2. Experiment section demonstrates state-of-the-art sub-degree accuracy and consistent improvements (up to ~30% absolute gains) across multiple major cross-view localization datasets, showing robustness and generalization.

**Weaknesses:**

1. In [CVPR 2020 - Where am I looking at? Joint Location and Orientation Estimation by Cross-View Matching], Shi et al. proposed cross-view image retrieval based on similarity between features encoded from ground panorama and polar-transformed aerial images, which is very similar to proposed method in terms of yaw alignment; proposed work claims disentanglement of yaw with the other 2 DoF, but limited experiment is performed and presented in supporting the claim, assuming $x$ and $y$ are accurate estimation is a strong assumption; hence would summarize for limited novelty and validation for proposed work.
2. Column-wise feature matching and yaw-bin scoring may incur significant computational overhead compared to end-to-end regression models, hence limited feasibility in real-world deployment.
3. in Eq.2, the $\|\|$ notation is not clearly indicated in terms of which kind of normalization.

**Questions:**

1. could the authors present more thorough experiment results on validating disentanglement of yaw improves 3 DoF pose accuracy with ablations?

---

> ### Author Response · Authors · 2025-11-21
>
> Dear Reviewer 7Qns, we would like to clarify the incorrect assumption underlying your concern and provide empirical results regarding the overhead.
>
> 1. ***Our method does not assume that $(x,y)$ is known or correct during yaw voting.***
> Instead, it computes a per-yaw score for ***all candidate $x$ and $y$ positions*** within the location noise, forming a 3D score bin. This allows the correct yaw to accumulate consistent votes across multiple 2D poses that align the radial line of the ground column with the matched BEV pixels. Please refer to our ***General Response*** for a concise explanation of how our method handles location uncertainty.
>
>     Our core contribution is the relative yaw formulation, which enables yaw estimation from a single linear structure even under location uncertainty. In contrast, the CVPR 2020 paper uses a polar transform that assumes the center pixel corresponds to the correct location, which makes it unsuitable for settings with uncertain locations. Their evaluation setup contains no location noise, differing from recent cross-view localization works and our setting, which explicitly evaluates under significant location perturbations.
>
> 2. Our method does not have significant overhead relative to existing cross-view localization methods. LAYS runs in 0.16s, compared to 0.26s for the two-staged FG2 pipeline. Replacing FG2’s rotation estimation with ours (Sec. 4.4.3) adds negligible overhead.
>
>      Cross-view localization could not be solved by a naive end-to-end regression model due to significant differences between views. For other cross-view methods, the overhead arises from point-wise or pixel-wise comparisons for a 3-DoF pose search space, which is comparable to our column-wise matching and yaw-bin scoring.
>
> 3. The notation || indicates absolute value, applied to the cosine similarity as noted in L209.
>
> Question 1 naturally stems from the incorrect assumption that our method treats the input location as correct and performs 1-DoF estimation. Our experiment in Section 4.4.3 demonstrates that yaw estimated from under location uncertainty can improve a 3-DoF estimator (FG2), highlighting the importance of isolating yaw estimation.
>
> We hope this clarification, especially regarding point 1, helps resolve the incorrect assumption and supports a fair reassessment of our work.

---

### Author Response · Authors · 2025-11-21
**General Response & Critical Clarification on Positional Uncertainty**

We thank all reviewers for their time and feedback, and are encouraged that N7WJ and 6GEP found our line-aligning formulation "novel," “intuitive,” and "practically effective.".

However, we must first clarify a ***key incorrect assumption*** (raised by 7Qns and snYM) that appears to underline critical concerns about our contribution, novelty, and experimental setup.

We clarify that: ***Our method does NOT assume a known position***; it explicitly operates under ***large location uncertainty*** (e.g., +-20m) during both training and evaluation. Concerns from 7Qns (“assuming x and y are accurate”) and snYM (“assuming ground-truth position is given”) are based on this incorrect premise.

Our method's novelty lies in estimating yaw without requiring an accurate (x, y), enabled by our  line-alignment formulation. ***The system is explicitly designed to handle location uncertainty, not to ignore it.***

We hope this clarification ***helps reassess our contributions*** under the correct problem setting. Our title and abstract explicitly state the ***location-uncertain setup***, and the introduction explains that “an accurate position is not commonly available,” motivating our formulation. The method and evaluation also detail the process of estimating yaw with unknown location, robustly accounting for the possibility of all locations within the noise range.

To summarize the key intuition (Fig. 1, Sec. 3.1, 3.4):

* A single column-to-line match defines a radial line of possible positions, not a point, that agrees on the correct yaw.
* Based on each pixel-wise match, for each 2D pose candidate (x, y) we compute a hypothesized yaw θ_est; incorrect poses produce inconsistent θ_est across pixels.
* Correct yaw yields consistent votes from linear-structure pixels over many (x, y), concentrating in the correct yaw bin of the 3D score volume S(θ, x, y).

Again, we do not assume (x, y) is known. We evaluate all candidate (x, y) positions through the exhaustive voting process, and the correct yaw is the one that yields consistent line-alignment across these uncertain locations, i.e., the yaw associated with the highest-scoring bin in the 3D score volume.

This noisy-location evaluation setup is explicitly described in the paper. As stated in L284: "we add perturbations to the ground truth poses in both position and orientation...", and the Table captions clearly indicate that location is noisy during evaluation.

This clarification is important because the incorrect assumption about a known location appears to cause several concerns, including the perception that our method resembles prior work.

We will update the Abstract and Introduction, and add clarifying remarks to make this central contribution more explicit. Detailed responses to individual comments will follow.

---

> ### Author Response · Authors · 2025-11-24
> **Paper Update: Remarks on Problem Setup and Additional Experiments**
>
> We have uploaded a revised version of the paper with changes highlighted in ***red***.
> * Remarks on the Problem Setup (Abstract, Sec. 1, 3.1, and 3.4)
> * Pitch and Roll Noise Robustness Experimental Results (Appendix)

---

### Comment · Area_Chair_H2zG · 2025-11-24

Dear Reviewer,

Thanks for taking the time to review this work. The authors have responded to your reviews. Can you please have a look at the rebuttal and discuss with the authors?

Best Regards,

AC

---

### Author Response · Authors · 2025-12-03
**Summary of Our Rebuttal and Final Remark (Part 1)**

We summarize the key points of our rebuttal. The initial low scores (7Qns & snYM) were driven by a **fundamental factual misunderstanding** of our problem setting. Correcting this misunderstanding removes the main concerns regarding novelty, validation, and practicality.

### **Resolution of the Critical Misunderstanding**
Reviewers 7Qns and snYM assumed that our method relies on ground-truth location (7Qns: "assuming x and y are accurate", snYM: "assuming ground-truth position is given"). They wrongly understood our method as an impractical method that assumes an unknown rotation with a known position, dramatically undermining the method’s contribution.

Our method (LAYS) is explicitly designed to estimate yaw **despite significant location uncertainty** (e.g., $\pm$20m). It operates by searching the full 3-DoF pose space to find the yaw that creates geometric consensus. This noisy-location setup is explicit in our Abstract, Introduction, and Evaluation (Sec 4.1).

Reviewer snYM acknowledged this misunderstanding, while 7Qns did not respond before the discussion was locked for reviewers.

### **Novelty: The Line-Aligning Formulation**
Novelty concerns, also raised by Reviewers 7Qns and snYM, stem from an incorrect understanding of our method’s setup and core components. Unlike prior works that require holistic image matching, our **Line-Aligning Formulation** enables yaw estimation from a single line correspondence, **free from ambiguity about the depth or distance** of features along the radial line.

**How LAYS works (Fig. 1, Sec 3.4):**
- **Radial Constraint:** Matches between a ground column and BEV pixels in a line define a **radial line** of possible positions and a yaw. Positionally, the score distributes along this line (inheriting uncertainty).
- **Yaw Voting:** For every candidate 2D pose $(x, y)$ in the noise range of each BEV pixel, we compute a hypothesized yaw $θ_{est}$ using the matching ground column's "Relative Yaw". Incorrect poses produce inconsistent, scattered $θ_{est}$ values.
- **Consensus Mechanism:** The correct yaw yields consistent votes from pixels along any linear structure (e.g., road, building edge) across the subset of $(x,y)$ locations that lies on the line. These votes concentrate in the correct yaw bin of our **3D score volume $S(\theta, x, y)$** for pose candidates $(x, y)$ in the line, effectively isolating the yaw despite the location uncertainty.

**Distinction from Prior Art:**
- vs. **Shi et al. (2020):** They rely on a polar transform of the aerial image to perform holistic 1D correlation, assuming that the center BEV pixel corresponds to the correct location. The approach does not consider locational uncertainty.
Our line-aligning formulation does not transform the view; instead, we focus on finding individual correspondence to accumulate evidence on which yaw is correct.
We explicitly model 3-DoF pose and reduce the difficulty of yaw estimation to a single-line correspondence search, robustly accounting for location uncertainty along the radial line (i.e., depth uncertainty in the ground).

- vs. **OrienterNet**: OrienterNet relies on template matching. It implicitly infers a dense 3D scene structure from the ground image with a polar transform and a depth template. It performs a 2D convolution (correlation) of the projected ground feature (Neural BEV) with the 2D map, matching the image as a whole for each possible pose.
In contrast, LAYS uses geometric consensus from a voting scheme. We do not hallucinate a scene layout. Instead, individual ground columns match to BEV pixels and "vote" for the yaw that satisfies the radial line constraint. The 3-DoF peak emerges not from image-to-image correlation, but from the geometric consensus of these independent votes.

---

> ### Author Response · Authors · 2025-12-03
> **Summary of Our Rebuttal and Final Remark (Part 2)**
>
> ### **Exceptional Performance in Realistic, Challenging Scenarios**
> Our method demonstrates exceptional performance in challenging scenarios where existing methods fail: **limited Field-of-View (FoV) inputs with significant noise in yaw.**
>
> | Method | Test 1 (Same Area) $<1^{\circ}$ | Test 1 (Same Area) $<2^{\circ}$ | Test 1 (Same Area) $<4^{\circ}$ | Test 2 (Cross Area) $<1^{\circ}$ | Test 2 (Cross Area) $<2^{\circ}$ | Test 2 (Cross Area) $<4^{\circ}$ |
> | :--- | :---: | :---: | :---: | :---: | :---: | :---: |
> | CCVPE* | 8.96 | $<$26.48 | $<$42.75 | 3.14 | $<$9.24 | $<$14.56 |
> | BoostAcc | 0.53 | 0.90 | 1.96 | 0.56 | 1.05 | 2.09 |
> | G2S | 0.58 | 1.17 | 2.17 | 0.54 | 0.82 | 1.64 |
> | FG2 | 1.62 | 3.15  | 6.47 | 1.62 | 2.96 | 6.13 |
> | **Ours** | **48.69** | **58.65** | **60.43** | **45.51** | **53.34** | **54.44** |
>
> (\* CCVPE paper evaluates unknown yaw setup, which shows 1, 3, 5 degree accuracy, we use it to show an upper bound for 1, 2, 4 degree accuracy.)
>
> From the evaluation on reviewer snYM’s request on the **KITTI dataset**, LAYS achieved **48.69%** and **45.51%** sub-degree accuracy for the same-area and cross-area test set with +-180$^\circ$ yaw noise and +-20m location noise. Prior methods, however, failed dramatically when we switched to a challenging +-180$^\circ$ yaw noise setup, even after fully retraining and tuning with their official open-source code, showing a similar trend with the MGL dataset result with +-180$^\circ$ noise. The best performing CCVPE achieves only ~9% sub-degree accuracy, and **most prior methods fail almost completely**.
>
> Prior methods were primarily tested only with a small yaw error (+-10$^\circ$), which makes yaw estimation trivial in the driving scenario, as vehicles follow the road lanes. **Prior methods do not withstand challenging scenarios in AR/MR**, where more orientation freedom exists (MGL dataset, our method achieves 34.81% sub-degree accuracy, second best CCVPE achieves 6.55%). They require an atypical camera with 360-degree visibility, like in VIGOR, to compare two views entirely and handle significant yaw noise.
>
> LAYS remains robust in these settings because it does not require a holistic match of the entire view; it builds consensus from available partial line-wise matches. We emphasize that **limited FoV (e.g., Ford, MGL, KITTI)** is the standard for most real-world vehicle, robotics, and MR cameras. LAYS's dominance here proves its value for general-purpose localization. Even for an omnidirectional camera in the VIGOR dataset (Table 3), LAYS achieves **23.67%** cross-area accuracy (<1$^\circ$), nearly **doubling** the performance of the next best method (12.39%).
>
> ### **Utility for the 3-DoF Localization**
> There were questions about whether LAYS improves full localization. We explicitly demonstrated this in **Section 4.4.3** and **Table 6**. By feeding our decoupled yaw estimate into a state-of-the-art 3-DoF method (FG2), we **reduced its translation error from 5.95m to 5.10m**, notably when FG2’s search space is explicitly reduced to 2-DoF. This proves LAYS is a valuable, performance-enhancing module for broader localization pipelines. Furthermore, because our method operates completely differently from typical 3-DoF localization methods, using a hybrid approach or feedback loop holds promise for further improving localization accuracy.
>
> ### **Considerations for Practical Use**
> **Robustness against Calibration Errors**
>
> Real-world systems often suffer from imperfect gravity alignment (pitch/roll errors). We evaluated LAYS on the MGL dataset by injecting random pitch and roll noise (±10$^\circ$) during training and testing. The method demonstrated high robustness, with the <1$^\circ$ yaw accuracy decreasing only marginally from 34.81% to 33.15%. This resilience stems from our relative yaw formulation. The estimation of our method is not affected by pitch error, as it uses radial line alignment of features rather than a strict ground-plane projection. The image encoder can adapt to roll variations via horizon cues.
>
> **Efficiency**
>
> LAYS achieves an inference latency of 0.16s, which is highly competitive compared to state-of-the-art 3-DoF methods (e.g., the two-stage FG2 pipeline requires 0.26s). While the exhaustive voting process spans a 3D search space, it is non-iterative, highly parallelizable, and computationally no more complex than existing cross-view localization methods.
>
> ### **Final Remark**
> With the fundamental misunderstanding resolved (acknowledged by snYM) and the method's unique capabilities validated on challenging benchmarks, LAYS represents a distinct advance focusing on yaw estimation in the cross-view localization problem. We respectfully request the AC to evaluate the paper based on this corrected understanding.

---

### Meta-Review · Area_Chair_Gqn1 · 2025-12-25

**Summary:**

This paper proposes LAYS, a voting-based line-alignment framework for yaw estimation in cross-view localization under large positional uncertainty. The key idea is that yaw can be isolated as a 1-DoF subproblem by leveraging geometric consensus along BEV–ground linear correspondences, even when the 2D location is highly ambiguous. The authors provide extensive experimental results on MGL, Ford, KITTI, and VIGOR, showing significant improvements over prior methods under extreme yaw noise.

While the rebuttal successfully clarified an important misunderstanding — namely that the method does not assume a known or accurate position — the paper still exhibits several fundamental limitations that prevent it from meeting the bar for ICLR. In particular, the contribution is narrowly scoped to yaw-only estimation, relies on strong structural assumptions, and is implemented through a large-scale discrete voting mechanism that is difficult to generalize, differentiate, or integrate into modern learning-based localization pipelines.

**Reviewer Concerns:**

Concerns addressed by the rebuttal:
1. The assumption that the method relies on ground-truth or accurate (𝑥,𝑦) positions was incorrect. The authors convincingly clarified that LAYS explicitly searches over the full 3-DoF pose space and is designed to operate under large location uncertainty.
2. Novelty concerns relative to Shi et al. (2020) and OrienterNet were partially alleviated once the problem setting was clarified.
3. The lack of KITTI evaluation was addressed by adding experiments under a high yaw-noise protocol.
4. Computational overhead concerns were mitigated by reporting a competitive runtime compared to existing pipelines.

Concerns that remain outstanding:
1. Strong reliance on linear-structure assumptions:
LAYS fundamentally assumes the presence of long, dominant linear structures (roads, building edges) to enable line-wise consensus. Its robustness in environments without such structures (open spaces, rural scenes) is not theoretically justified or empirically evaluated.
2. Algorithmic scalability and differentiability:
The approach relies on an exhaustive, discrete voting scheme over (θ,x,y), with non-differentiable yaw binning and multiple heuristic hyperparameters. This design hinders end-to-end learning, limits scalability to larger BEV grids, and raises concerns about integration with modern differentiable localization frameworks.
3. Nature of the novelty:
The main innovation lies in reformulating point-wise matching into line-consensus voting. This is primarily an engineering-level reformulation rather than a new learning principle, probabilistic model, or inference paradigm.

**Reviewer Scores:**

Reviewer 7Qns: Likely to move from Reject to Borderline / Weak Reject after the clarification, but would still maintain a negative stance due to limited scope and structural assumptions.
Reviewer snYM: Would likely raise the score after acknowledging the positional-uncertainty clarification, but remain unconvinced by the narrow contribution and limited generality.
Other reviewers: Likely to maintain their original assessments.

---

### Decision · Program_Chairs · 2026-01-26

Reject